# The Role of Soluble LAG3 and Soluble Immune Checkpoints Profile in Advanced Head and Neck Cancer: A Pilot Study

**DOI:** 10.3390/jpm11070651

**Published:** 2021-07-10

**Authors:** Andrea Botticelli, Ilaria Grazia Zizzari, Simone Scagnoli, Giulia Pomati, Lidia Strigari, Alessio Cirillo, Bruna Cerbelli, Alessandra Di Filippo, Chiara Napoletano, Fabio Scirocchi, Aurelia Rughetti, Marianna Nuti, Silvia Mezi, Paolo Marchetti

**Affiliations:** 1Department of Clinical and Molecular Oncology, University of Rome “Sapienza”, 00185 Rome, Italy; andrea.botticelli@uniroma1.it (A.B.); paolo.marchetti@uniroma1.it (P.M.); 2Laboratory of Tumor Immunology and Cell Therapy, Department of Experimental Medicine, Policlinico Umberto I, University of Rome “Sapienza”, 00161 Rome, Italy; ilaria.zizzari@uniroma1.it (I.G.Z.); alessandra.difilippo@uniroma1.it (A.D.F.); chiara.napoletano@uniroma1.it (C.N.); fabio.scirocchi@uniroma1.it (F.S.); aurelia.rughetti@uniroma1.it (A.R.); marianna.nuti@uniroma1.it (M.N.); 3Department of Medical and Surgical Sciences and Translational Medicine, University of Rome “Sapienza”, 00185 Rome, Italy; 4Department of Molecular Medicine, University of Rome “Sapienza”, 00161 Rome, Italy; giulia.pomati@uniroma1.it; 5Medical Physics Unit, “S. Orsola-Malpighi” Hospital, 40138 Bologna, Italy; lidia.strigari@aosp.bo.it; 6Department of Radiological, Oncological and Anatomo-Pathological Science, University of Rome “Sapienza”, 00185 Rome, Italy; alessio.cirillo@uniroma1.it (A.C.); bruna.cerbelli@uniroma1.it (B.C.); silvia.mezi@uniroma1.it (S.M.)

**Keywords:** LAG3, soluble immune checkpoints, head and neck cancer, immunotherapy

## Abstract

Unresectable recurrent and/or metastatic head and neck squamous cell carcinoma (R/M HNSCC) has a very poor prognosis. Soluble immune checkpoints (sICs) are circulating proteins that result from the alternative splicing of membrane proteins and can modulate the immune response to cancer cells. The aim of our pilot study was to determine the possible role of a comprehensive evaluation of sICs in the classification of prognosis and response to treatment in patients with advanced disease. We evaluated several sICs (CD137, CTLA-4, PD-1, PD-L1, PD-L2, TIM3, LAG3, GITR, HVEM, BTLA, IDO, CD80, CD27, and CD28) from peripheral blood at baseline and investigated the association with clinical characteristics and outcomes. A high baseline soluble LAG3 (sLAG3 > 377 pg/mL) resulted in an association with poor PFS and OS (*p* = 0.047 and *p* = 0.003, respectively). Moreover, sLAG3 emerged as an independent prognostic factor using an MVA (*p* = 0.005). The evaluation of sICs, in particular sLAG3, may be relevant for identifying patients with worse prognoses, or resistance to treatments, and may lead to the development of novel targeted strategies.

## 1. Introduction

Head and neck squamous cell carcinomas (HNSCCs) represent globally the sixth most common cancer with 890,000 cases/year and 450,000 death/year in 2018 [1]. Unresectable recurrent and/or metastatic (R/M) HNSCC has a very poor prognosis, with median survival of one year from diagnosis [2]. In the past, upfront treatment options included platinum-based combination regimens with the addition of cetuximab, an anti EGFR monoclonal antibody [3,4]. In a second-line setting, taxanes and methotrexate were the most common options, though neither of these drugs showed a benefit in overall survival (OS) [5]. Recently, immunotherapy has profoundly changed the management of R/M HNSCC both in first line treatment and in platinum-resistant disease. The phase III trial Checkmate 141, which included 361 patients with R/M HNSCC who progressed after platinum-based regimens were randomized to receive nivolumab or mono-chemotherapy based on the investigator’s choice, led to the approval of nivolumab in this disease setting regardless of the PD-L1 expression [6]. The phase III trial Keynote 048 evaluated the upfront treatment pembrolizumab, an anti PD-1 agent, as monotherapy or in association with platinum-based chemotherapy versus chemotherapy plus placebo regimen [7] in 882 patients with R/M HNSCC. Pembrolizumab improved OS in patients with combined positive score (CPS) ≥ 20 (14.9 vs. 10.7 months) and in CPS ≥ 1 population (12.3 vs. 10.3 months) compared to the EXTREME regimen. Pembrolizumab plus chemotherapy improved OS in the total population (13 vs. 10.7 months), in a subgroup with CPS ≥ 20 (14.7 vs. 11 months), and in CPS ≥ 1 (13.6 vs. 10.4 months) compared to the EXTREME regimen [8]. These results sanctioned the introduction of pembrolizumab in the first-line setting in CPS ≥ 1 disease. However, only a limited subset of patients with HNSCC gain a long-term benefit from immune checkpoint inhibitors (ICIs), underlining the critical role of patients’ choice in undergoing the right treatment.

The HNSCC cancer microenvironment is characterized by a high inflammatory component and a high level of tumor-infiltrating lymphocytes (TILs). The programmed death 1 (PD-1), and programmed death ligand 1 (PD-L1) axis is involved in the initial escape process from the immune system, maintenance, and progression of HNSCC and it is the main target of immune checkpoint inhibitors (ICIs) [9,10]. PD-L1 is a dynamic biomarker, but is considered an imperfect predictor of response or resistance to immunotherapy as it cannot faithfully reflect the complex immune profile of patients. Furthermore, in clinical trials, there was no homogeneity in the choice of PD-L1 determination method.

The possible predictive role of PD-L1 has been extensively investigated in several solid tumors, including renal cancer, non-small cell lung cancer (NSCLC), and melanoma [11,12,13,14]. However, the prognostic and predictive role of PD-L1 varies widely among tumor types and the location of disease, emphasizing the existence of different immune mechanisms as the bases of carcinogenesis and tumor progression [15]. Consequently, there is a need to identify an adequate method of studying the immune system in order to fully understand its complexity.

Several soluble immune molecules play a role in immune system regulation. Soluble immune checkpoints (sICs) are circulating proteins, resulting from the alternative splicing of membrane proteins, which can modulate immune response to cancer cells [16]. The determination of sICs present in the patient may reflect their immune status and effectively predict changes following the different treatments received. The disease stage (lymph nodes involvement and metastasis) as well as the types of treatments received (radiotherapy, chemotherapy, and surgery) can alter the patient’s immune profile and impact the efficacy of immunotherapy. sICs can provide important information in order to stratify patients and personalize immunotherapy approaches.

The aim of our pilot study was to determine the role of comprehensive sICs evaluation as a possible prognostic and predictive factor in patients with advanced disease.

## 2. Materials and Methods

### 2.1. Clinical Data

This pilot study included patients with advanced or metastatic HNSCC who started treatment with chemotherapy or immunotherapy between December 2018 and June 2020. Baseline staging was performed according to the AJCC 8th edition of TNM system [17] with contrast-enhanced computed tomography (CT) and contrast-enhanced magnetic resonance imaging (MRI). Age, sex, baseline Eastern Cooperative Oncology Group performance status (ECOG PS), treatment received, comorbidities, tumor histology, tobacco smoking, and alcohol abuse were the baseline data collected. We included patients 18 years or older, with histologically confirmed HNSCC of the oral cavity, oropharynx, or larynx, and who were not eligible for curative local therapies (surgery and/or radiotherapy) or had metastatic disease spread. Patients were ECOG PS ≤ 1 with permissive hematopoietic, hepatic, and renal function; fit for chemotherapy or fit for immunotherapy after progression of disease to a first line treatment with platinum-based chemotherapy; and able to provide a signed informed consent. Patients with PS > 2, a non-squamous, or a rare histology, or patients who were not eligible for the selected treatments were not included in the study.

Patients were treated with chemotherapy or with nivolumab. Response to treatment was evaluated with a CT scan or an MR and clinical examination every 3 months as per clinical practice and according to RECIST 1.1 [18] for patients treated with chemotherapy and iRECIST [19] for patients treated with immunotherapy. Progression-free survival (PFS) was defined as the period from the first administration of treatment until the first documented tumor progression or death from any cause, expressed in months. Overall survival (OS) was defined as the period from treatment commencement to death or last follow up available, expressed in months. Data were collected anonymously into a specific database. The protocol approval of the local ethics committee was obtained [CE 4212].

### 2.2. Immunomonitoring

Peripheral blood samples were collected at baseline (T0) before the first administration of any treatment. After centrifugation, serum samples were collected and stored at −80 °C until use. The soluble immune checkpoints (sICs) were evaluated through a multiplex assay using the Human Immuno-Oncology Checkpoint 14-plex ProcartaPlex Panel 1 (catalog number EPX14A-15803-901) (Thermo Fischer Scientific, Waltham, MA, USA). We evaluated the following sICs: CTLA4, CD137, PD-L1, PD-L2, PD-1, LAG3, BTLA, IDO, TIM3, CD80, GITR, HVEM, CD27 and CD28. For each patient, 50 μL of serum was used and added to a 96-well plate together with a mixture of magnetic beads coated with antibody according to the manufacturer’s instructions. A biotinylated detection antibody was added to the plate and then bound to phycoerythrin (PE)-conjugated streptavidin. Samples were analyzed using Luminex 200 platform (BioPlex, Bio-Rad, Hercules, CA, USA). Data are expressed in pg/mL and were analyzed with Bio-Plex Manager Software.

### 2.3. Statistical Analysis

Continuous variables are presented in a descriptive table. Mean and range values were calculated. Boxplots were calculated for each soluble factor according to the adopted treatment. The boxes enclose the interquartile range, and the horizontal line is the median in each case. The whiskers extend to the highest and lowest value still within 1.5× the interquartile range, and circles represent the outliers. The matrix of Pearson’s correlations for all the pairs of soluble factors was calculated using the R-package Hmisc and shown using the R-package PerformanceAnalytics.

The impact of soluble factors on overall survival (OS) and progression-free survival (PFS) was analyzed by both univariate (UVA) and multivariate analyses (MVA). Concerning UVA, patients’ OS and PFS were analyzed using the Kaplan–Meier method and log-rank tests. Prognostic clinical and pathological variables such as age, sex, baseline Eastern Cooperative Oncology Group (ECOG) performance status (PS), stage, tumor location, treatment received, and sICs values deemed of potential relevance in the univariate analysis (corresponding to a cut-off of *p* < 0.100 at the UVA) were included in the MVA analysis. Considering the number of subjects in our pilot study (i.e., 23 patients) a maximum of 2 variables were included in the MVA according to the rule of thumb. A *p* < 0.050 was considered statistically significant. R-package software was used to perform statistical analyses.

## 3. Results

In this pilot study, we evaluated sICs on 23 patients with advanced or metastatic HNSCC treated with chemotherapy or anti PD-1 antibody. Clinical and pathological features are shown in Table 1.

The median age was 62 years (40–71 years). Ten patients were treated with chemotherapy and thirteen with a nivolumab 240 mg flat dose every 2 weeks. Fourteen patients had a baseline PS 1 (61%). The most common primary site was the oral cavity (43.5%). Ten patients had a locally advanced cT4 and cN2 or cN3 while thirteen had a metastatic disease. The values of sICs at baseline is reported in Table 2.

Several sICs showed a significant correlation at baseline. A Pearson’s correlation > 0.8 indicates a strong correlation between variables (e.g., sLAG3 values are statistically significantly associated and directly correlated with CTLA4, PD-L1, GITR, CD80, and IDO values with a correlation value of 0.9, 0.85, 0.87, 0.78, and 0.77, respectively; *p* < 0.001; Figure 1).

Using the UVA, the treatment type (i.e., nivolumab, *p* = 0.024) and median sLAG3 > 377 pg/mL (Figure 2, *p* = 0.047) were significantly associated with lower PFS (Table 3). Regarding clinical and pathological features, age (≥62 years), sex, and baseline ECOG PS were not significantly associated with lower PFS (*p* = 0.513, *p* = 0.401, and *p* = 0.118, respectively) or OS (*p* = 0.755, *p* = 0.919, and *p* = 0.900, respectively).

Only median sLAG > 377 pg/mL (Figure 3, *p* = 0.001) was significantly associated with lower OS (Table 3). Furthermore, patients with high sLAG3 showed a significant lower PFS (*p* = 0.047) and OS (*p* = 0.001) compared to those with low baseline value (Figure 2 and Figure 3).

At the MVA, the two variables included in the models of PFS and OS were the median baseline sLAG > 377 pg/mL and treatment type, having a *p*-value <0.100. Baseline sLAG > 377 pg/mL was found to be a significant prognostic factor of OS (*p* = 0.005), while it was associated to PFS as a trend (*p* = 0.067, Table 4).

## 4. Discussion

Immune checkpoints (ICs) are molecules that regulate the modulation of the immune system and have been investigated as relevant factors in treating cancer. Currently, the inhibition of ICs represents a pillar in cancer immunotherapy, including treatment of HNSCC. Despite the long-term benefit, a large number of patients experience resistance to ICIs and rapid disease progression.

In this scenario, the identification of new biomarkers and new combination strategies is crucial to improve patient outcomes. Several circulating immune factors that can be evaluated allow a better understanding of a patient’s immune response. Among these, the most important are cytokines and soluble checkpoints. Cytokines are proteins, peptides, or glycoproteins, secreted by specific cells, which modulate the action of other specific immune cells [20]. Soluble receptors and ligands seem to derive from the cleavage of membrane-bound proteins or by mRNA expression and can be found free in the plasma [16]. The study of circulating factors such as soluble immune checkpoints (sICs), cytokines, and chemokines, as well as the detection of exhausted circulating lymphocytes, are topics of growing interest, representing new challenges in the field of immuno-oncology. Soluble immune checkpoints may represent immune parameters to evaluate the dynamic behavior of the immune system in patients during treatment, avoiding the need for multiple biopsies to study the changes in tumor microenvironment [21].

A recently published trial demonstrated a strong correlation between serum level of soluble cytotoxic T-lymphocyte antigen 4 (sCTLA4) and both response rate and survival in patients with metastatic melanoma treated with ipilimumab. In particular, high levels of sCTLA4 were associated with ipilimumab response and improved OS [22]. Moreover, in an exploratory study, both sPDL1 and sCTLA4 were correlated with poor outcome in patients with metastatic renal clear cell carcinoma treated with tyrosine kinase inhibitors (TKI) [23]. Lymphocyte-activation gene-3 (LAG-3) is a crucial immune checkpoint, involved in the regulation of T cell activation and proliferation in cancer patients and expressed on lymphocytes membrane [24]. Exhausted or dysfunctional T cells excessively exposed to infection or cancer antigens have a higher expression of LAG3 compared to naïve or activated ones [25]. LAG3 is also expressed on natural killer (NK) cells, dendritic cells (DCs), and B lymphocytes [26]. LAG-3 binds with major histocompatibility complex class II (MHC II) with a higher affinity than CD4+ [27]. Moreover, LAG-3 can bind with the LSECtin, a member of the C-type lectin receptor superfamily, typically exposed on tumor cell membranes, and with the fibrinogen like protein 1 (FGL-1), which is part of the fibrinogen protein superfamily [28]. Several trials have reported that LAG-3 regulates inhibitory effect on T cells synergizing with the PD-1/PD-L1 axis [29]. Finally, a trial with BMS-986016, a LAG-3 monoclonal antibody, alone or in combination with nivolumab for pretreated advanced solid tumors, is currently ongoing.

In HNSCC, LAG3 overexpression on tumor tissue has been associated with larger tumor size, extensive lymph node involvement, and higher tumor grade. Moreover, LAG3 levels are higher in cancer cells derived from metastatic nodes and recurrent disease compared to those from the initial tumor. Finally, patients with higher LAG3 expression have a poor prognosis [30]. For this reason, LAG3 is considered as one of the most promising avenues in the treatment of HNSCC, and several combination strategies are currently under evaluation [31].

Soluble LAG3 (sLAG3) is released by shedding at the cell membrane and provides a further immune regulation in peripheral blood and TME, performing different functions from the membrane counterpart. sLAG3 can impair monocyte differentiation, producing APC with reduced immune activation characteristics [32]. The role of sLAG3 has not been clarified. Soluble LAG3 has been evaluated in different studies, demonstrating a predictive value of prognosis in several solid tumors, including gastric and breast cancer [33,34], primary liver cancer [35], non-small cell lung cancer (NSCLC), and clear cell carcinoma [36]. In NSCLC in particular, low sLAG3 was associated with extended locally advanced or metastatic disease [36]. In breast and gastric cancer, detectable levels of sLAG3 were associated with improved prognosis [33,34]. In vivo experiments on mice showed that sLAG3 is able to promote cell-mediated immune response, increasing the expression of IFN-γ and IL-12 [33]. These data suggest that further investigation of sLAG3 as a prognostic or predictive biomarker is required. However, considering the important role of LAG3 in suppressing immune response and generating resistance to ICIs in HNSCC, it is possible to hypothesize an association with worse prognosis.

These findings suggest that sLAG3 may become the next object of cancer therapies to be combined in a synergistic way with anti PD-1 agents. In the near future, the identification of a circulating immune profile may become a flexible approach able to highlight the mechanisms of resistance to immunotherapy as well as to direct the clinician towards the personalization of treatment through promising combination strategies [37].

In our pilot study, sLAG3 seems to be associated with poor prognosis in HNSCC. Using UVA, a high sLAG value was associated with patients who had a significantly shorter PFS (*p*= 0.047) and OS (*p* = 0.0013). At MVA, sLAG3 seems to be associated with prognosis regardless of disease stage and treatment (*p* 0.0051; Table 4).

Our results show that the basal values of many sICs are associated with each other. It is therefore possible to hypothesize that patients have a complex circulating immune profile rather than single, separate, circulating factors. The basal values of sLAG3 are significantly associated and directly correlated with CTLA4, PD-L1, GITR, CD80, and IDO (correlation value 0.9, 0.85, 0.87, 0.78, and 0.77 respectively; *p* < 0.001; Figure 1). A higher sLAG3 value corresponds to a higher value of all the other cited sICs, highlighting the presence of a comprehensive immune suppressive profile.

The introduction of immunotherapy in the treatment of solid tumors led us to the possibility of modulating the immune system. In this context, it is becoming clear that the cure for cancer should involve not only the treatment of cancer cells, but also a re-modulation of the cancer-patient network (CPN). The CPN is the result of the complicated interactions between cancer, cancer microenvironment, immune system, and the host. The soluble profile (cytokine, sICs, and chemokines) is the expression of the dynamic plasticity of the CPN, in which we may find the best biomarkers and targets. From this perspective, the CPN and its modulation represent the main hub for our strategies.

## 5. Conclusions

Considering our preliminary results, the evaluation of sICs, in particular sLAG3, may be relevant for identifying patients with poor prognoses and resistance to treatments. Moreover, the identification of circulating factors and their interpretation in a network approach may lead to a more precise and targeted immunotherapy.

## Figures and Tables

**Figure 1 jpm-11-00651-f001:**
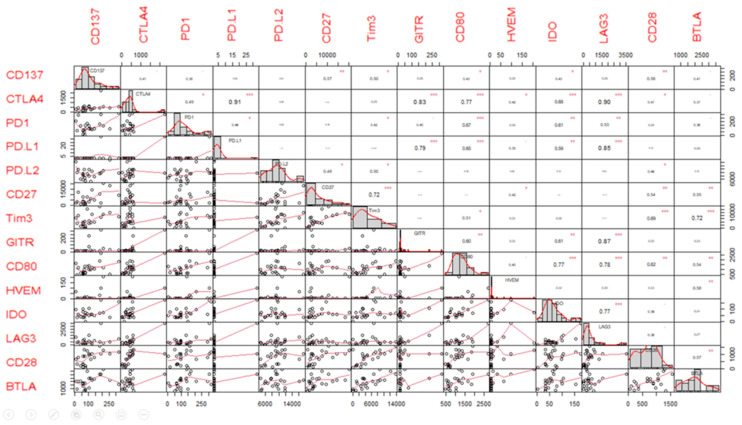
Baseline sICs values and Pearson’s correlation matrix for soluble pairs. This allows for the identification of the distribution of each variable (on the diagonal), the bivariate scatter plots between variables with a fitted line (on the bottom of the diagonal), and the correlations between investigated variables and the significance level of the Pearson’s correlation test. The Pearson’s correlation matrix is for all the pairs of soluble factors reported in Table 2. The distribution of each variable, expressed in pg/mL, is shown on the diagonal. On the bottom of the diagonal, the bivariate scatter plots with a fitted line are displayed. On the top of the diagonal, the value of the correlation plus the significance level of the correlation test as stars are reported. The significance levels are associated with a symbol i.e., the *p*-values of <0.001, <0.01, and <0.05 are indicated as “***”, “**”, and “*”, respectively. *p*-values indicating a trend (*p* < 0.100) are indicated as “.”.

**Figure 2 jpm-11-00651-f002:**
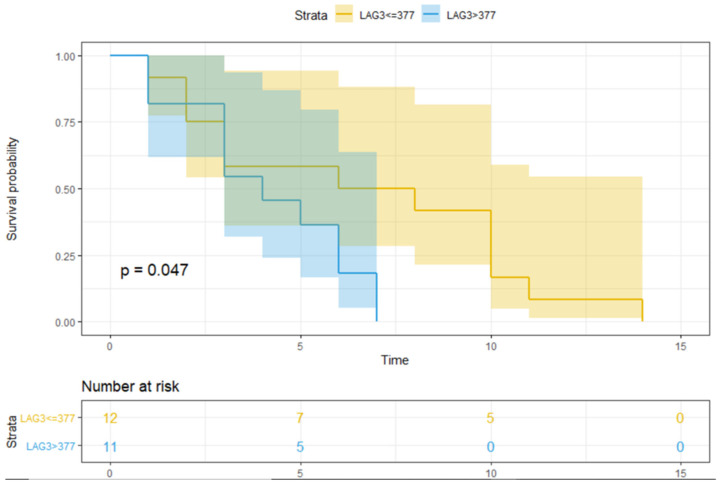
The PFS over time according to sLAG3 value. Figure 2 shows the PFS over time (expressed in months after the start of treatment) in patients with/without an LAG3 value higher than the median. The number of patients at risk is also indicated. PFS refers to progression-free survival; 377 is the median value of LAG3, expressed in pg/mL; the blue line represents OS in patients with sLAG3 > 377 pg/mL, and the yellow line represents OS in patients with sLAG3 ≤ 377 pg/mL; the shaded bands represent the confidence intervals at each time point. The table under the figure indicates the number of subjects in each group (i.e., sLAG3 > 377 or ≤377 pg/mL) at baseline (0) or at 5, 10, or 15 months after the beginning of treatment. *p* = *p*-value; significant *p*-values are indicated in bold.

**Figure 3 jpm-11-00651-f003:**
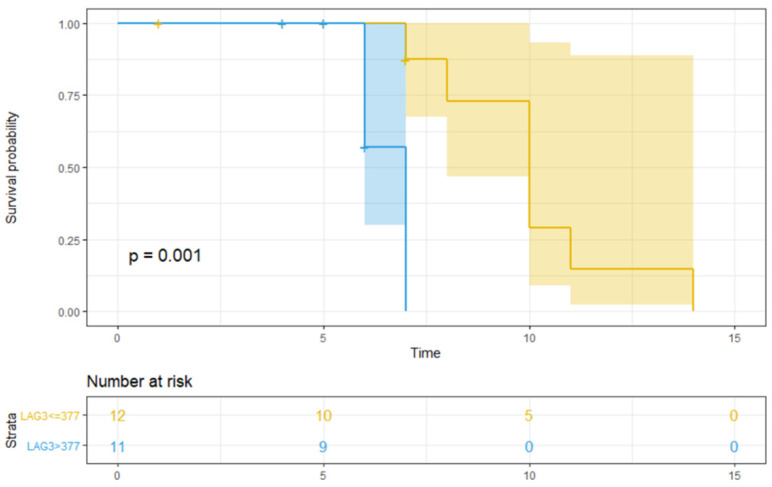
The OS over time according to baseline sLAG3 value. Figure 3 shows the OS over time (expressed in months after the start of treatment) in patients with/without an LAG3 value higher than the median. The number of patients at risk is also indicated. OS refers to overall survival; 377 is the median value of LAG3, expressed in pg/mL; the blue line represents OS in patients with sLAG3 > 377 pg/mL, and the yellow line represents OS in patients with sLAG3 ≤ 377 pg/mL; the shaded bands represent the confidence intervals at each time point. The table under the figure indicates the number of subjects in each group (i.e., sLAG3 > 377 or ≤377 pg/mL) at baseline (0) or at 5, 10, or 15 months after the beginning of treatment. *p* = *p*-value; significant *p*-values are indicated in bold.

**Table 1 jpm-11-00651-t001:** Clinical and pathological characteristics.

	Patients N (%)
Age (years)	
Median Age (range)	62 (40–71)
Sex	
Male	16 (70.0)
Female	7 (30.0)
Performance Status	
0	9 (39.0)
1	14 (61.0)
Stage	
Locally advanced	10 (43.5)
Metastatic	13 (56.5)
Tumor Location	
Oral cavity	10 (43.5)
Oropharynx	7 (30.4)
Larynx	6 (26.1)
Histology	
Squamous Cell Carcinoma	23 (100)
Treatment	
First Line chemotherapy	10 (43.5)
Nivolumab	13 (56.5)

**Table 2 jpm-11-00651-t002:** sICs baseline values.

Variable	Median (pg/mL)	Range (pg/mL)
BTLA	2035	648–3716
CD137	76	9–320
CD27	3211	316–20,449
CD28	885	60–1580
CD80	1019	461–2692
CTLA4	402	4–2269
GITR	13	7–318
HVEM	6	6–206
IDO	56	2–174
LAG3	377	6–3546
PD-L1	2	2–33
PD-L2	9257	4551–17,313
PD-1	98	4–318
Tim3	4028	36–13,776

**Table 3 jpm-11-00651-t003:** *p*-values of univariate analysis of PFS and OS according to each sICs.

sICI Median Value (pg/mL)	PFS	OS
BTLA > 2035	0.700	1.000
CD137 > 76	0.700	0.600
CD27 > 3211	0.800	0.900
CD28 > 885	0.700	0.400
CD80 > 1019	0.700	0.700
CTLA4 > 402	0.900	0.400
GITR > 13	0.700	0.900
HVEM > 6	0.300	0.300
IDO > 56	1.000	0.700
**LAG3 > 377**	**0.047**	**0.001**
PD-L1 > 2	0.800	0.500
PD-L2 > 9257	0.600	0.300
PD-1 > 98	0.700	0.900
Tim3 > 4028	0.700	0.600

The cutoff of investigated groups is the median value of sICs expressed in pg/mL. PFS: progression-free survival; OS: overall survival; variables significantly associated to PFS or OS are indicated in bold *p* ≤ 0.05.

**Table 4 jpm-11-00651-t004:** MVA of PFS and OS.

Endpoint	Variable	β	Exp (β)	95% CI	*p*-Value of Model
PFS	LAG3 > 377 pg/mL	0.9163	2.5	0.9152 to 6.8291	0.067
OS	LAG3 > 377 pg/mL	2.4707	11.8313	1.4118 to 99.1472	**0.005**

The MVA baseline value of sLAG3 is confirmed as an independent prognostic factor. PFS: progression-free survival; OS: overall survival; MVA: multivariate analysis; CI: confidence interval; Significant *p*-values are indicated in bold.

## Data Availability

Data available on request due to restrictions eg privacy or ethical. The data presented in this study are available on request from the corresponding author.

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
