# Peer review of "The Role of Soluble LAG3 and Soluble Immune Checkpoints Profile in Advanced Head and Neck Cancer: A Pilot Study"

_jpm, 2021, doi:10.3390/jpm11070651_

Round 1

Reviewer 1 Report

The paper is well written.  The methods are adequately described.  The conclusions follow from the results.  The only (minor) point is to increase the text within Figure 1 for better readability.

Reviewer 2 Report

In the submitted manuscript Botticelli et al. have shown the results of a pilot study on the utility of soluble immune checkpoints as prognostic biomarkers for advanced head and neck cancer.

Although authors clearly stated that this is a pilot study, this study and manuscript has so many flaws that it cannot be accepted for publication.

Just to name a few...

1) English language style is too informal, there are too many colloquial phrases and imprecisions. It definitively has to undergo English proofreading. Especially "2.3 Statistical analysis" subsection because it is incomprehensible. It is more description of data presentation than statistical analysis methods.

2) Authors stated that "p<0.05 was considered statistically significant" while already in 'Abstract' they presented a result with p=0.05 as significant.

3) Throughout the manuscript P-values were arbitrary presented with 1, 3 or 4 decimals. Completely incomprehensible why.

4) Figures should not have explanations beneath and footnotes. All should be explained in a figure legend.

5) Figure 1 is a mess, unreadeable and incomprehensible. In addition, if you set a statistical significance threshold to P<0.05, why emphasize correlation coefficients with P<0.1?!

6) Survival curves are incomprehensible, need more descriptions (what was a unit of time, what present color shades, etc.).

7) Units of measurements of sICs were nowhere mentioned or presented in a text and tables.

8) Data in Table 2 were presented by arbitrary values. It should be with mean, SD, median and range. What should IQR present in particular?!

9) At many points the reference is missing, e.g, for the 8th edition of TNM system (is it UICC or AJCC?!), RECIST 1.1 and iRECIST, after text in lines 65-68, etc.

10) Word "influence" should be avoided because you have not proven how sICs really affect patients' survival.

11) UVA and MVA were not properly described nor presented. Which all (clinico-pathological) variables were included in them?! Since you have only 23 patients, by well known rule of thumb (https://en.wikipedia.org/wiki/One_in_ten_rule), you can include only two variables in MVA!

To summarize, although it is a pilot study, it does not (and in fact cannot) follow standard rules for reporting a tumor marker prognostic study (https://pubmed.ncbi.nlm.nih.gov/22675273/).

Author Response

please see the attachemnt

Reviewer 3 Report

Nice paper to read. the topic is very interesting. the chalenge of biomarkers (prognostic) is the future in oncology. 

I was wondering see the correlation between sLAG3 and the immunotherapy group? what about the tumor P16 marker?  

The CPS wasn not a standard during the study inclusion, but I think this score could be usefull to test with sLAG3.

line 70: no > non small ...

line 101 : fist > first

Round 2

Reviewer 2 Report

Authors have substantially improved the quality of their manuscript and satisfactorily answered to all my comments.

I have just noticed that the name of right column in Table 1 has unnecessary "23", that percentage in line 153 has decimal comma instead of point, and that p-values in lines 175 and 176 have 4 decimals while all other have 3.
